# Anxiety, PTSD, and stressors in medical students during the initial peak of the COVID-19 pandemic

Carmen M. Lee[1]*, Marianne Juarez[1], Guenevere Rae[2], Lee Jones[1], Robert M. Rodriguez[1], John A. Davis[1], Megan Boysen-Osborn[3], Kathleen J. Kashima[4], N. Kevin Krane[2], Nicholas Kman[5], Jodi M. Langsfeld[6], Aaron J. Harries[1]

1 University of California San Francisco School of Medicine, San Francisco, California, United States of America, 2 Tulane University School of Medicine, New Orleans, Louisiana, United States of America, 3 University of California Irvine School of Medicine, Orange, California, United States of America, 4 University of Illinois College of Medicine, Chicago, Illinois, United States of America, 5 Ohio State College of Medicine, Columbus, Ohio, United States of America, 6 Donald and Barbara Zucker School of Medicine at Hofstra/Northwell, Hempstead, New York, United States of America

* carmen.lee2@ucsf.edu

## Abstract

### Purpose

To assess psychological effects of the initial peak phase of the COVID-19 pandemic on United States (US) medical students in clinical training to anticipate sequelae and prepare for future outbreaks.

### Methods

Authors emailed a cross-sectional survey in April-May, 2020 to students in clinical training years at six US medical schools which included validated General Anxiety Disorder (GAD-7) and Primary Care-PTSD (PC-PTSD-5) screening tools, and asked students about pandemic-related stress and specific concerns. Authors used quantitative and thematic analysis to present results.

### Results

**Of 2511 eligible students, 741 responded (29.5%)**. Most students (84.1%) reported at least "somewhat" increased levels of stress and anxiety related to the pandemic. On the GAD-7, 34.3% showed mild, 16.1% moderate, and 9.5% severe anxiety symptoms, with 39.6% demonstrating no/minimal symptoms. One quarter (25.4%) screened positive for PTSD risk symptoms. Top concerns of students chosen from a pre-populated list included inadequate COVID-19 testing, undiagnosed or asymptomatic spread and racial or other disparities in the pandemic. In thematic analysis, students' reactions to removal from clinical learning included: understanding the need to conserve PPE (32.2%), a desire to help (27.7%), worry over infectious risk to others (25.4%) and self (21.2%), and lost learning opportunities (22.5%). Female students were significantly more likely to report anxiety and PTSD risk symptoms. Asian students had a greater risk of moderate anxiety and those

**Data Availability Statement:** The data for this study can be found at https://www.kaggle.com/carmenmarielee/medstudent-mental-health-2020/.

**Funding:** The author(s) received no specific funding for this work.

**Competing interests:** The authors have declared that no competing interests exist.

underrepresented in medicine (UIM) had greater risk of moderate and severe anxiety symptoms compared to white students.

## Conclusions

During the initial peak phase of COVID-19, over 60% of US medical students screened positive for pandemic-related anxiety and one quarter were at risk for PTSD. Female and UIM students were significantly more affected. Medical schools should consider broad support of students, and targeted outreach to female and UIM students.

## Introduction

The profound mental health effects of the COVID-19 pandemic on healthcare providers have been well-documented [1], however less is known about the effects on medical students. Studies prior to the pandemic have demonstrated that medical student psychological distress varies by year, and is greater in clinical years; there is evidence that medical students have higher levels of anxiety and depression compared with the general population and age-matched peers [2, 3]. Further, medical students are less likely to access mental health care due to stigma and concerns regarding career progression [4]. Given that students who do access mental health resources have shown improvement in suicidality, emotional distress and mental illness, identification of situations and at-risk students is critical [4].

A mid-2020 Centers for Disease Control (CDC) survey of the general population during the pandemic showed increased rates of depression, anxiety, and suicidal ideation especially in younger adults [5]. Personal and academic stressors related to COVID-19, including the removal of many medical students from clinical rotations during the initial peak phase of the pandemic [6], may have increased stress on this already vulnerable population [7]. Medical students experienced major disruptions to their education as a result of the pandemic, and most experienced increased burnout and stress [8].

Pandemics, natural disasters, and other loss-of-life events are associated with increased anxiety, depression, and PTSD among providers and students. For example, medical students in Malaysia and Saudi Arabia experienced increased anxiety during the response to SARS-CoV1 and MERS [9, 10]. Medical students had significant PTSD and depression during Hurricane Katrina [11], and provider PTSD was well-described after the September 11 attacks [12].

There is global evidence that the COVID-19 pandemic has also posed unique stresses to medical students, highlighting the need for further study in the US. Medical students in Canada and Australia have shown significant mental health and anxiety effects during the COVID-19 pandemic [7, 13]. In studies from Poland, India, Morocco, and Saudi Arabia, the pandemic anxiety burden on female students was significantly greater [14–17]. Iran and the United Arab Emirates also showed greater burden amongst female students and students in clinical training years [18–20]. A social media study in the US suggests rates of anxiety and depression among medical students may have increased during the pandemic [21], but there has not yet been an attempt to use census sampling to estimate prevalence of anxiety or PTSD in this population.

Our study investigates the mental health effects of the initial peak phase of the COVID-19 pandemic on clinical medical students. We assess anxiety and PTSD using validated screening tools and identify specific stressors with a thematic analysis of student reactions to removal from clinical rotations. We hope this study will better elucidate the significant mental health

impact of the pandemic on this vulnerable population, and also provide a foundation for medical schools to anticipate and address student needs going forward.

## Methods

We emailed a cross-sectional survey to all medical students in clinical training at six medical schools across the United States during the initial peak phase of the COVID-19 pandemic between 4/20/20 and 5/25/20. We intentionally recruited participating schools in order to represent the four different regions of the country and account for varying COVID-19 prevalence: University of California San Francisco School of Medicine (San Francisco, CA), University of California Irvine School of Medicine (Orange, CA), Tulane University School of Medicine (New Orleans, LA), University of Illinois College of Medicine (Chicago, Peoria, Rockford, and Urbana, IL), Ohio State University College of Medicine (Columbus, OH), and Zucker School of Medicine at Hofstra/Northwell (Hempstead, NY). We defined study participants as any medical student involved in clinical training, including graduation class years of 2020, 2021, and 2022 depending on the curricular schedule of each school and the intended graduation timeline of the student. Details of the eligible population at each school are given in S1 Table. We excluded students who had not started clinical rotations. The study was deemed exempt by the respective Institutional Review Boards.

### Survey instrument

The survey instrument included two validated mental health screening tools, the Generalized Anxiety Disorder-7 (GAD-7) and the Primary Care PTSD Screen DSM-5 (PC-PTSD-5). These were supplemented with questions to elicit students' pandemic-related concerns drawn from a published survey studying the pandemic effects on Emergency Medicine physicians [1]. The final survey included 29 Likert, multiple choice, and free response questions. The survey underwent iterative review drawing on authors' experience in educational program evaluation and psychometric measures and was piloted with a group of five medical students for clarity and completion time. After review by medical school deans, the survey was distributed via email with three reminders from student representatives. Results were collected anonymously.

### Scoring and analysis

The GAD-7 scoring system was validated in the outpatient setting and uses a 21-point score to identify mild (score of 5–9), moderate (10–14), or severe (15 +) anxiety symptoms [22]. The PC-PTSD-5 asks 5 yes/no questions; 3 or more "yes" answers was considered a positive screen as previously validated [23]. Perceived stress and concerns were evaluated using a 7-point unipolar scale from "not at all" (1) to "somewhat" (4) and "extremely" (7). (Survey Instrument, S1 File). Both the GAD-7 and the PC-PTSD-5 have been used in the US and internationally to measure the effects of the COVID-19 pandemic on healthcare workers [24]. In stratified analysis, UIM students were defined in accordance with the American Association of Medical Colleges (AAMC) as those whose racial or ethnic identities are underrepresented in medicine as compared to the general US population [25].

We used Qualtrics to manage the survey data, hosted by the University of California, San Francisco, and performed data analysis using STATA v16.1 (Stata Corp, College Station, TX). Respondent characteristics and key responses are summarized as raw counts, frequency percent, medians, and interquartile ranges (IQR). We assessed differences between groups for significance using Pearson's chi-squared testing with a significance level of 0.05.

## Thematic analysis

The survey included an open response question that asked participants to explain the reasons underlying their reactions to having a clinical rotation cancelled due to COVID-19. We conducted a thematic analysis to identify themes in participants' responses. Two raters (one fourth-year medical student and one Associate Professor with experience in qualitative analysis) independently read the responses and reflected on their meaning to generate a list of categories from each "complete thought" identified. The raters discussed their interpretations of each category and combined ones that were very similar to generate a final set used to independently re-code the dataset. We assessed inter-rater reliability using a random subset of 30 responses by unweighted kappa and evaluated the distribution of responses using Chi-squared analysis.

# Results

We received 741 responses out of 2511 students contacted for the survey (response rate 29.5%). Most respondents identified as female (n = 443, 63.9%), with 35.1% (n = 243) male and 1.6% (n = 9) expressing a different gender identity. The majority of students (n = 557, 80.0%) lived with a partner, roommate, or family member. A small percentage lived with someone over the age of 70 (n = 20, 2.9%) or someone at otherwise higher risk for complications from COVID-19 (n = 43, 6.2%) (Table 1). Students had limited patient contact at the time of the study, with 93.7% (n = 694) reporting that they were not involved in direct patient care.

## Anxiety and PTSD risk

Most students (n = 619, 84.1%) perceived at least somewhat increased stress due to the pandemic. On the GAD-7, a majority of students scored in the range for mild (n = 253, 34.4%), moderate (n = 119, 16.1%), or severe symptoms of anxiety (n = 70, 9.5%). No/minimal symptoms were reported for 39.6% (n = 292) of students. On the PC-PTSD-5, 25.4% (n = 188) of students met criteria for a positive PTSD screen with a score of 3 or greater, with 11.6% (n = 86) scoring 4 or greater (median score 1 out of 5, IQR 0,3). A summary of responses to these scales by item is shown in S2 Table. Students reported getting an average of 7.43 hours of sleep per night during the pandemic, with 45.8% (n = 338) describing this as more than, and 37.3% (n = 275) the same as pre-pandemic averages.

Anxiety and PTSD scores were stratified by key metrics (Table 2). Students whose gender identity included female had significantly higher levels of anxiety symptoms and PTSD risk (p<0.01). When white-identifying students were compared with those who identified as Asian or UIM, no significant differences were observed in perceived stress or risk for PTSD, although these groups differed on GAD-7 scores (p<0.01). UIM students had greater rates of moderate and severe anxiety than white students, and Asian students had greater moderate anxiety only. Older students had higher rates of severe but lower moderate and mild anxiety than younger students (p = 0.02). Both perceived stress (p<0.01) and GAD-7 anxiety (p<0.01) differed significantly by graduation year, with students in the class of 2020 reporting lower levels of anxiety on both scales. Students' living arrangement was associated with significant GAD-7 (p = 0.02) and PC-PTSD-5 (p = 0.045) differences, with those who lived alone or with someone at high risk for severe COVID-19 demonstrating greater moderate-severe anxiety burden and positive screening for PTSD.

## Pandemic-related concerns

When asked to rate their agreement with specific statements about concerns related to the pandemic, students were most worried about inadequate testing in their community,

**Table 1. Demographics.**

|  | *n* (%) |
|---|---|
| **Age** |  |
| 20–24 | 87 (12.5%) |
| 25–29 | 493 (70.9%) |
| 30–34 | 98 (14.1%) |
| 35–59 | 14 (2.0%) |
| 40–44 | 1 (0.1%) |
| 45+ | 2 (0.3%) |
| **Gender Identity**\* |  |
| Includes Male | 243 (35.1%) |
| Includes Female | 443 (63.9%) |
| Trans Male | 1 (0.1%) |
| Trans Female | 0 (0.0%) |
| Genderqueer/Nonbinary | 6 (1.2%) |
| Other | 2 (0.3%) |
| **Race/Ethnicity**\* |  |
| African-American | 42 (6.1%) |
| Asian | 199 (29.1%) |
| Hispanic/Latinx | 77 (11.2%) |
| Native American/American Indian | 5 (0.7%) |
| Native Hawaiian/Pacific Islander | 1 (0.1%) |
| White | 408 (59.6%) |
| Other | 36 (5.3%) |
| **Living Environment**\* |  |
| Alone | 92 (13.2%) |
| Roommate | 247 (35.5%) |
| Partner | 281 (40.4%) |
| Someone Under Age 18 | 28 (4.0%) |
| Someone Over Age 70 | 20 (2.9%) |
| My Parent(s) | 120 (17.2%) |
| Friend(s) | 61 (8.8%) |
| Someone Else at Risk for Severe COVID | 43 (6.2%) |
| Other | 44 (5.2%) |
| **Graduation Year**† |  |
| 2020 | 193 (27.8%) |
| 2021 | 372 (53.5%) |
| 2022 | 120 (18.7%) |
| **US Region** |  |
| West | 290 (41.8%) |
| Midwest | 114 (16.4%) |
| South | 176 (25.4%) |
| East | 110 (15.9%) |

Percentages were calculated out of responses to each demographic question.

\*numbers do not sum to 100 because questions allowed more than 1 response.

†excluding any planned time off.

**Table 2. Stratification of anxiety results by gender, ethnicity, and other factors.**

| | n | Perceived Stress | | GAD-7 Anxiety | | | | PC-PTSD-5 | |
| --- | --- | --- | --- | --- | --- | --- | --- | --- | --- |
| | | At Least "Somewhat" | p* | Mild | Moderate | Severe | p* | Screen Positive | p* |
| **Gender** | | | | | | | | | |
| Includes female | 443 | 398 (89.8%) | <**0.001** | 156 (34.9%) | 80 (18.3%) | 48 (11.0%) | **0.001** | 144 (32.5%) | <**0.001** |
| Includes male | 244 | 179 (73.4%) | | 73 (30.4%) | 31 (12.9%) | 16 (6.7%) | | 41 (16.8%) | |
| **Race/Ethnicity†** | | | | | | | | | |
| White (non-Latinx) | 359 | 295 (82.2%) | 0.27 | 126 (35.9%) | 54 (15.4%) | 29 (8.3%) | **0.009** | 89 (24.8%) | 0.39 |
| Asian | 189 | 162 (85.7%) | | 68 (36.4%) | 35 (18.7%) | 10 (5.3%) | | 56 (29.6%) | |
| Underrepresented | 137 | 120 (87.6%) | | 37 (27.2%) | 22 (16.2%) | 24 (17.6%) | | 40 (29.2%) | |
| **Age** | | | | | | | | | |
| 20–24 | 87 | 77 (88.5%) | 0.11 | 30 (34.9%) | 24 (27.9%) | 7 (8.1%) | **0.024** | 26 (29.9%) | 0.26 |
| 25–29 | 493 | 418 (84.8%) | | 167 (34.4%) | 75 (15.5%) | 45 (9.3%) | | 137 (27.8%) | |
| 30+ | 115 | 90 (78.3%) | | 35 (31.0%) | 12 (10.6%) | 13 (11.5%) | | 24 (20.9%) | |
| **Graduation Year** | | | | | | | | | |
| 2020 | 193 | 148 (76.7%) | **0.004** | 54 (28.3%) | 27 (14.1%) | 10 (5.2%) | <**0.001** | 52 (26.9%) | 0.65 |
| 2021 | 372 | 323 (86.6%) | | 124 (34.1%) | 56 (15.4%) | 41 (11.3%) | | 96 (25.8%) | |
| 2022 | 130 | 114 (87.7%) | | 54 (41.9%) | 28 (21.7%) | 14 (10.9%) | | 39 (30.0%) | |
| **Living Arrangement‡** | | | | | | | | | |
| Lives Alone | 85 | 75 (88.2%) | 0.43 | 24 (28.2%) | 16 (18.8%) | 15 (17.6%) | **0.021** | 30 (35.3%) | **0.045** |
| Lives with High-Risk | 54 | 47 (87.0%) | | 20 (38.5%) | 13 (25.0%) | 5 (9.6%) | | 19 (35.2%) | |
| Lives with Other | 557 | 464 (83.3%) | | 188 (34.3%) | 82 (15.0%) | 45 (8.2%) | | 138 (24.8%) | |
| **Regional COVID˚** | | | | | | | | | |
| High Prevalence | 289 | 240 (83.0%) | 0.53 | 96 (33.6%) | 44 (15.4%) | 21 (7.3%) | **0.034** | 65 (22.5%) | 0.091 |
| Medium Prevalence | 113 | 99 (87.6%) | | 31 (27.9%) | 17 (15.3%) | 20 (18.0%) | | 35 (31.0%) | |
| Low Prevalence | 292 | 246 (84.2%) | | 105 (36.6%) | 49 (17.1%) | 24 (8.4%) | | 86 (29.5%) | |

[†]Race/ethnicity groups consisted of: any student selecting an underrepresented in medicine (UIM) identity including African-American, Hispanic/Latinx, Native American/American Indian, Native Hawaiian/Pacific Islander; any student selecting Asian identity without UIM; and any student selecting white identity alone.

[‡]Students living with someone at high risk resided with someone over 70 years old or otherwise at elevated risk for complications of COVID-19 (such as with comorbidities). Students living with others included those who reported roommates, partners, children under 18, parents, or others not in the high risk category.

[˚]Regional COVID prevalence in April-May 2020 was used to determine groups as high (NY, LA), medium (CA), and low (OH, IL) prevalence.

[*]All significance testing was performed using Pearson's chi-squared test with significance level set at 0.05.

undiagnosed or asymptomatic community spread, and racial or other disparities in outcomes from COVID-19 (median 6; IQR 5,7) (Fig 1). Students were less concerned about exposure to violence while sheltering at home (median 1; IQR 1,1), access to food or necessities (2; 1,3) or being exposed to the virus while at school (2; 1,4). Approximately one third of students (n = 279, 37.7%) said they had taken precautions beyond those recommended to the general public as a result of their risk as a healthcare worker, including avoiding friends and family, changing and showering immediately on arrival home, and serving as an example in their community by masking and social distancing.

## Thematic analysis

Students' reactions to removal from clinical rotations were mixed, with a majority (n = 341, 59.7%) selecting two or more conflicting reactions on the survey. In the open response portion of the question, 311 students elaborated on their reasoning. Using qualitative analysis, fourteen themes emerged amongst these results (interrater reliability, kappa = 0.69) (Table 3). The most

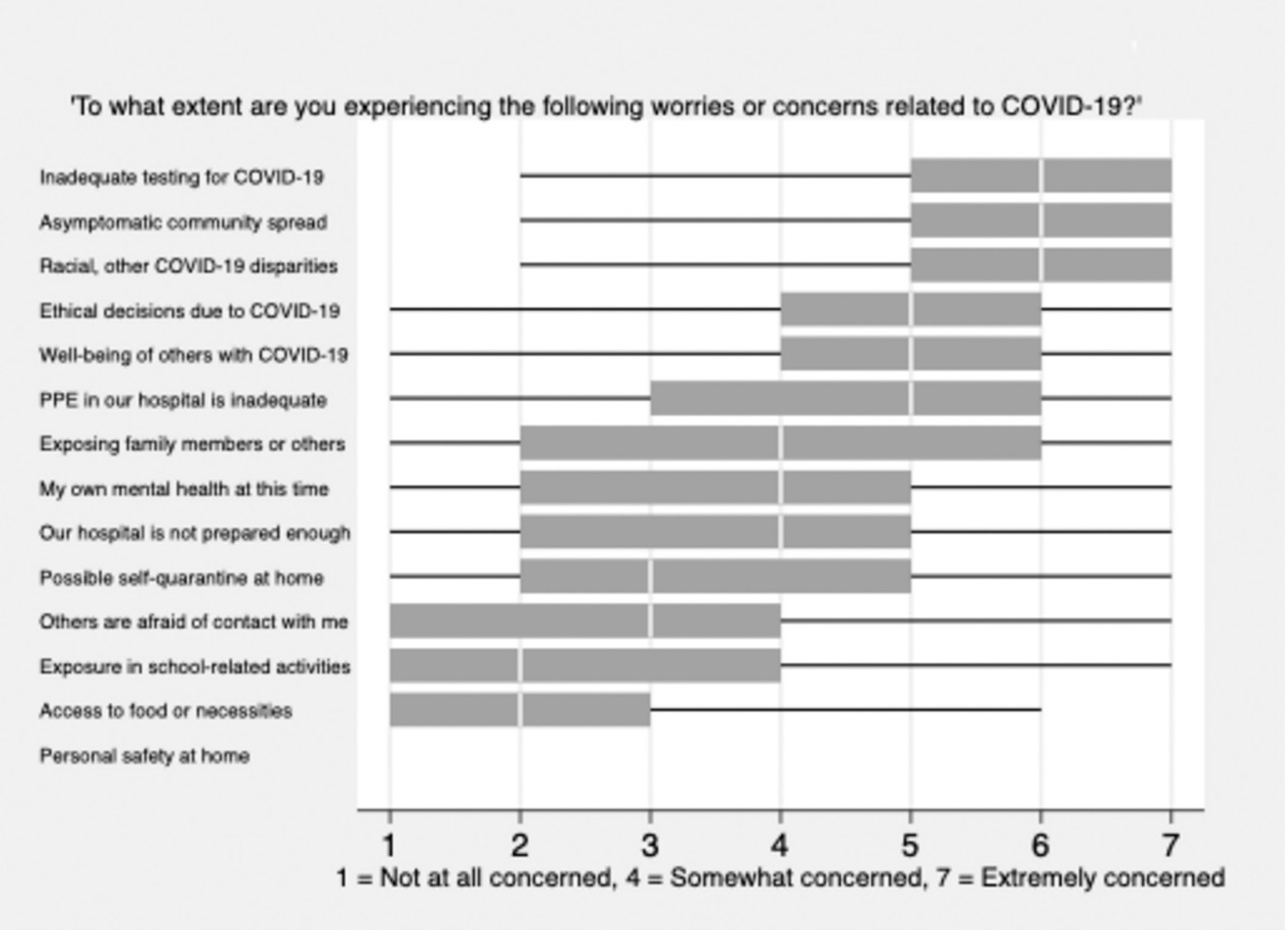

**Fig 1. Specific concerns of senior medical students.**

frequent themes described concern about conservation of PPE (n = 100, 32.2%), a desire to help with the pandemic (n = 86, 27.7%), risk of infection to others (n = 79, 25.4%) and to self (n = 66, 21.2%), and the loss of clinical learning opportunities (n = 70, 22.5%). Responses often described tension between these concerns:

> "My first reaction was relief because I have profound anxiety about unwittingly exposing patients to harm by being in the clinic. I do feel like it is important to continue my training in order to be competent to care for patients in the future, but feel conflicted about the tension between my obligation to future and current patients."

## Discussion

In this study we have found that medical students show adverse mental health effects due to the pandemic, with some groups, including women and underrepresented minorities, carrying greater mental health burden. We captured a cross-section during the initial peak phase of the

**Table 3. Reactions to removal from clinical environments.**

| | |
|---|---|
| Conservation of PPE *100* (32.2%) | "We didn't have adequate PPE for licensed providers, so I felt it appropriate to not include students in patient care when we would use additional PPE without enabling a commensurate increase in patient care capabilities." |
| Desire to Help with Pandemic *86* (27.7%) | "I was really looking forward to being back in the hospital to care for patients. I wanted to be able to help in this time of need. It sucks feeling like I can't contribute in the way I'd hope to at this time." |
| Decreased Risk to Others *79* (25.4%) | "In the setting of a pandemic, medical students act as additional vectors for infection. With no healthcare worker shortage our presence was worsening containment of spread. I did not believe a few weeks of my education was [sic] worth people's lives." |
| Loss of Learning Opportunity *70* (22.5%) | "I have always learned better from the patient rather than from a textbook and felt this was actually an incredible learning opportunity—once every one hundred years to see the care of these people." |
| Decreased Personal Risk *66* (21.2%) | "I am personally very high risk and was concerned for my own health when we weren't initially suspended from clinical rotations. That being said, my physician did clear me for certain settings and I would have liked to help more." |
| Feeling Unwanted or Useless *49* (15.8%) | "I think that deep down, I knew that I wasn't an essential member of the team, but I still felt that I was contributing to patient care. Being banned from the hospital reinforced that I was not, and although I know that it was the safe decision and done in the name of patient care, I was still disappointed to be sidelined when my future colleagues are still doing the necessary work." |
| Career or Residency Concern *39* (12.5%) | "Unfortunately I was on the rotation that I am planning to pursue as a career. While I felt relieved that I wouldn't be serving as a potential vector to infect patients, I was also disappointed that my rotation was cut short and am worried about what this will mean for my residency applications going forward." |
| Decreased Burden to Teams *35* (11.3%) | "I felt that we would be taking valuable time and energy away from doctors and residents who would need to teach and supervise us during a time when those valuable resources should be reserved for treating COVID patients" |
| Time for Other Priorities *23* (7.4%) | "I have had time to focus on clinical knowledge and feel I will be better prepared when I get to go back. I believe my 'job' at this time is to further my medical knowledge, so when I can come back I can be as helpful in the clinical environment as possible." |
| Modified Teaching + Learning Environment *23* (7.4%) | "Clinical responsibilities had decreased substantially in the days leading to the decision to have students remain at home. During this time, I felt that my clinical learning had decreased substantially and I felt less involved in patient care." |
| Stressful Institutional Response or Messaging *19* (6.1%) | "I feel like we've received mixed messaging from the school. On the one hand we are deemed essential and then all of my rotations have been cancelled. This has been frustrating." |
| Limited Impact of Removal *17* (5.5%) | "It wasn't in an area I was interested in and I felt the risk outweighed the potential educational value." |
| No Reasoning Cited *18* (5.8%) | "I was disappointed, but I also feel like I understand and that it was necessary." |
| Moral Objection to Helping *11* (3.5%) | "I do feel guilty that I am unable to help during this time, but expecting students (who are paying for an educational experience) to work without pay or benefits is morally inappropriate, and I respected my school's decision to keep us out of the hospital." |

pandemic in the United States: at that time, a majority of students screened positive for at least mild anxiety in response to the pandemic, one quarter showed moderate to severe anxiety, and one quarter screened positive for PTSD. Because most students had been removed from clinical rotations, the mental health effects occurred independent of whether students were involved in direct patient care.

Students' specific pandemic-related concerns centered on risks to their colleagues and patients; testing and asymptomatic transmission in their communities; and racial and other disparities in the effects of the pandemic. Of note, our results were collected prior to the widespread reporting of racial and ethnic disparities in COVID-19 by the CDC in July 2020 and our survey closed prior to protests related to racial injustice after the death of George Floyd, indicating concern for local disparities before they captured national attention. Students were less concerned about personal exposure risk due to the pandemic, yet more concerned about exposing others through their healthcare work. The pattern of anxiety described in these statements shows that students most strongly felt concerns related to their clinical teams, their hospital systems, and their communities, even after they had been removed from in-person clinical learning.

Students' conflicting concerns were revealed by thematic analysis of reactions to removal from clinical rotations and should help to inform the implementation of clinical learning during the ongoing pandemic (Table 3). Some concerns, such as access to PPE for themselves and the entire team, may not recur as planning and preparedness improve. Others, such as worry about the burden posed to clinical teams by students or institutional messaging that increased stress with mixed or unclear rationale, offer opportunities for interventions that help students process these conflicts. The tension students describe between lost learning opportunities and their own and others' safety is not likely to resolve until more robust vaccinations are possible and may persist even then.

These conflicting concerns—for community, colleagues, and self—may increase pressure on students to choose between what is best for their patients, their health, or their early careers. Even when this choice was removed, as in the unique window we studied, their answers often describe stress caused by these conflicts. The idea of feeling unwanted or useless identifies challenges to professional identity formation during the salient clinical years. This study demonstrates objective effects on mental health as students are pulled in different directions by this unprecedented educational environment. The complex landscape of students' concerns highlights that support may need to be individualized and adapted to circumstance.

The stratification of our results indicates that burdens are higher for some subgroups. Women had higher anxiety and PTSD risk, similar to previously documented greater risk for anxiety in female medical students and female healthcare workers during this pandemic [1, 26]. The increased risk for anxiety observed in UIM and Asian medical students also echoes trends observed in providers [27]. It is difficult to speculate about the reasons for these differences, but these consistent disparities in the mental health effects of the pandemic deserve further exploration and support. These groups may benefit from tailored outreach, such as through multicultural resource centers. Medical schools should also consider proactive contact with all students to identify those showing signs of severe strain in this later phase of the pandemic. Especially concerning was the greater risk for moderate-severe anxiety and PTSD in students living alone, or with high-risk individuals, given the erosion of pre-pandemic social support. Specific needs of these groups should be elicited, but personal outreach could in and of itself help strengthen students' ties to the school community.

The primary limitation of our study is the low response rate. Other surveys conducted during the pandemic have also demonstrated lower than expected response rates [5], suggesting that broader disruptions are decreasing the yield of traditional techniques. Bias may have been introduced by fewer respondents to the survey: in addition to the class of 2020, who graduated early at half of the schools surveyed, there may be large groups of students whose circumstances prevented them from responding. As a result, it is possible that the mental health burden is even greater than we have reported here. The scope of this study focused on anxiety and PTSD specifically; there are likely other mental health burdens that were not assessed, such as

suicidality. With substantial evidence of higher prevalence of depression during the pandemic in general populations, this topic should be explored further in medical students.

## Conclusions

Medical students are screening positive for anxiety symptoms and are at risk for PTSD in the setting of the COVID-19 pandemic. Specifically, female and UIM students are at a significantly higher risk of these mental health effects related to the pandemic when compared to their peers. Given previous data suggesting medical students are already a vulnerable population with higher levels of anxiety, depression and psychological distress, they may be at higher risk for adverse outcomes or long-term sequelae in response to these ongoing stresses. Students feel conflicted regarding their need to continue clinical learning as healthcare providers and the need to maintain safety for themselves and those around them. We have employed two validated screening tools to assess for anxiety symptoms and the risk of PTSD that may be helpful to other institutions in addressing the mental health effects of the COVID-19 pandemic. Our future medical workforce will undoubtedly continue to feel the effects of this pandemic for years to come.

## Supporting information

**S1 File. Complete survey instrument.**
(DOCX)

**S1 Table. Eligible clinical student populations by medical school.**
(DOCX)

**S2 Table. Descriptive statistics for anxiety and PTSD scales.**
(DOCX)

## Acknowledgments

The authors wish to thank Sara Ackerman, MPH, PhD and Nancy Hills, PhD for their support and feedback, and Jill Barr-Walker, MPH, MS for her help with our review of the literature.

**Please note**: All authors made substantial contributions to this study and met the specific conditions listed in the guidelines for authorship. Given that the study was multicenter, we included twelve authors in order to capture a broad sampling of medical students from across the country at six geographically targeted medical schools. All authors have read and approved the manuscript.

## Author Contributions

**Conceptualization:** Carmen M. Lee, Marianne Juarez, Lee Jones, Robert M. Rodriguez, John A. Davis, Aaron J. Harries.

**Data curation:** Carmen M. Lee, Megan Boysen-Osborn, Kathleen J. Kashima, N. Kevin Krane, Nicholas Kman, Jodi M. Langsfeld.

**Formal analysis:** Carmen M. Lee, Marianne Juarez, Guenevere Rae, Robert M. Rodriguez, Aaron J. Harries.

**Investigation:** Carmen M. Lee, Marianne Juarez, Guenevere Rae, Lee Jones, John A. Davis, Megan Boysen-Osborn, Kathleen J. Kashima, N. Kevin Krane, Nicholas Kman, Jodi M. Langsfeld, Aaron J. Harries.

**Methodology:** Carmen M. Lee, Marianne Juarez, Guenevere Rae, Lee Jones, Robert M. Rodriguez, John A. Davis, Aaron J. Harries.

**Project administration:** Carmen M. Lee, Marianne Juarez, Lee Jones, John A. Davis, Megan Boysen-Osborn, Kathleen J. Kashima, N. Kevin Krane, Nicholas Kman, Jodi M. Langsfeld, Aaron J. Harries.

**Resources:** Marianne Juarez, Lee Jones, Robert M. Rodriguez, John A. Davis, Megan Boysen-Osborn, Kathleen J. Kashima, N. Kevin Krane, Nicholas Kman, Jodi M. Langsfeld, Aaron J. Harries.

**Software:** Carmen M. Lee.

**Supervision:** Marianne Juarez, Lee Jones, Robert M. Rodriguez, John A. Davis, Aaron J. Harries.

**Validation:** Carmen M. Lee, Marianne Juarez, Guenevere Rae, Lee Jones, Robert M. Rodriguez, Aaron J. Harries.

**Visualization:** Carmen M. Lee, Marianne Juarez, Guenevere Rae.

**Writing – original draft:** Carmen M. Lee, Marianne Juarez, Guenevere Rae, Aaron J. Harries.

**Writing – review & editing:** Carmen M. Lee, Marianne Juarez, Guenevere Rae, Lee Jones, Robert M. Rodriguez, John A. Davis, Megan Boysen-Osborn, Kathleen J. Kashima, N. Kevin Krane, Nicholas Kman, Jodi M. Langsfeld, Aaron J. Harries.

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
