## [Decision Letter · Decision Letter 0]

8 Jul 2021

Anxiety, PTSD, and stressors in medical students during the initial peak of the COVID-19 pandemic

PONE-D-21-17823

Dear Dr. Lee,

We’re pleased to inform you that your manuscript has been judged scientifically suitable for publication and will be formally accepted for publication once it meets all outstanding technical requirements.

Kind regards,

Arista Lahiri

Academic Editor

PLOS ONE

Additional Editor Comments (optional):

Reviewers' comments:

Reviewer's Responses to Questions

**Comments to the Author**

1. Is the manuscript technically sound, and do the data support the conclusions?

Reviewer #1: Yes

Reviewer #2: Partly

2. Has the statistical analysis been performed appropriately and rigorously? 

Reviewer #1: Yes

Reviewer #2: Yes

3. Have the authors made all data underlying the findings in their manuscript fully available?

Reviewer #1: Yes

Reviewer #2: Yes

4. Is the manuscript presented in an intelligible fashion and written in standard English?

Reviewer #1: Yes

Reviewer #2: No

5. Review Comments to the Author

Reviewer #1: The article has been revised as per recommendations and can be accepted for publication. In the point to point response file, all queries of the reviewers have been addressed appropriately. Therefore this manuscript can be published as original article.

Reviewer #2: Novelty of this study is questionable. However, much effort has been spend to conduct the study, which is appreciable. But there are better scales compared to the ones used for the purpose they were meant too.

6. PLOS authors have the option to publish the peer review history of their article (what does this mean?). If published, this will include your full peer review and any attached files.

Reviewer #1: No

Reviewer #2: No

---

## [Editor Report · Acceptance letter]

19 Jul 2021

PONE-D-21-17823 

Anxiety, PTSD, and stressors in medical students during the initial peak of the COVID-19 pandemic 

Dear Dr. Lee:

I'm pleased to inform you that your manuscript has been deemed suitable for publication in PLOS ONE. Congratulations! Your manuscript is now with our production department. 

Kind regards, 

on behalf of

Dr. Arista Lahiri 

Academic Editor

PLOS ONE